# Regulation of foliar salicylic acid at the heading stage enhances the grain 2-acetyl-1-pyrroline content and yield in fragrant rice

Siren Cheng[1¤☯], Xueqing Wang[2☯], Likai Zheng[2], Jinyue Liao[3], Qingqing Li[4], Yong Ren[1]*

1 College of Biological Sciences and Technology, Yili Normal University, Yining, Xinjiang, PR China,
2 College of Smart Agriculture, YuLin Normal University, YuLin, Guangxi, PR China, 3 College of Biology and Pharmacy, YuLin Normal University, YuLin, Guangxi, PR China, 4 College of Agriculture, Yangtze University, Jingzhou, Hubei, PR China

☯ These authors contributed equally to this work.
¤ Jiefang West Road No.448, Yining, Xinjiang Uygur Autonomous Region, P.R. China
* renyong20190076@163.com

## Abstract

Despite numerous studies on salicylic acid (SA) and its functions, research regarding the regulation of foliar-applied SA at the heading stage in fragrant rice remains scarce. In this study, a pot experiment was conducted using *Meixiangzhan* and *Yuxiangyou-zhan* cultivars. Exogenous SA was applied at four concentrations, namely 0 (CK), 0.1 mM (SA1), 0.5 mM (SA5), and 1 mM (SA10), during the heading stage to assess its effects on fragrant rice production. The results demonstrated that foliar SA application significantly enhanced the grain 2-acetyl-1-pyrroline (2-AP) content. The SA5 treatment yielded the highest values, with increases of 68.2% and 22.7% for *Meixiangzhan* and *Yuxiangyouzhan*, respectively. Moreover, SA application led to improvements in the activity of proline dehydrogenase and the contents of proline, pyrroline-5-carboxylic acid, and pyrroline. Regarding yield, SA application resulted in an increase, with both cultivars achieving a maximum under the SA5 treatment. This was mainly attributed to the enhancement of 1000-grain weight and seed-setting rate, as well as the improved antioxidant properties, particularly the activity of antioxidant enzymes. Principal component analysis revealed distinct differences between the control and treatment groups. Correlation analysis indicated that yield and yield – related traits were both directly and indirectly associated with the 2-AP content, and were regulated by antioxidant oxidase activity and proline content. Overall, the 0.5 mM SA treatment represents an effective strategy for synergistically improving both the yield and aroma of fragrant rice.

## Introduction

Rice, as one of the paramount cereal crops, has been cultivated historically and consumed bulkily in China [1]. Fragrant rice, in which 2-acetyl-1-pyrroline (2-AP)

**Data availability statement:** All relevant data are within the paper and its Supporting Information files.

**Funding:** This research was supported by the Natural Science Foundation of Guangxi Zhuang Autonomous Region (grant number: 2021GXNSFBA196084 (S.C.) and 2023GXNSFBA026307(Y.R.)), the Middle-aged and Young Teachers' Basic Ability Promotion Project of Guangxi (grant number: 2023KY0609 (Y.R.)).

**Competing interests:** The authors have declared that no competing interests exist.

has been identified as the most important contributor to aroma among more than 200 volatile substances, is more desirable to Chinese farmers because of its strong scent, fine taste, and high nutritional value [2,3]. However, compared with nonfragrant rice, the lower yield induced by weaker resistance to various stresses has been the main factor limiting the large-scale planting of fragrant rice [3,4]. Hence, enhancing the antioxidant properties, 2-AP content, and yield productivity of fragrant rice is a primary objective for improving its quality and value.

In addition to changes in gene expression, agricultural management practices, and external growth conditions, the growth process, yield formation, and 2-AP accumulation of fragrant rice are also modulated by the application of several plant growth regulators (PGRs) [4–6]. For example, growth development and 2-AP biosynthesis in seedlings of fragrant rice could be facilitated by increasing the contents of methylglyoxal, pyrroline, proline, and pyrroline-5-carboxylic acid (P5C); the enzyme activities of proline dehydrogenase and proline dehydrogenase (PDH); and the gene expression of *ProDH*, *P5CS2*, and *DAO4* when plants are sprayed with 20–40 µM trans-zeatin [7]. Zhang et al. and Xing et al. reported that foliar application of 1–2 µM methyl jasmonate (MeJA) and 100–120 mg·L$^{-1}$ paclobutrazol at the heading stage could increase the 2-AP content of grains [8,9]. Moreover, improvements in seedling growth, heavy metal tolerance, antioxidant activity, and even the grain quality and output of fragrant rice under cadmium and lead toxicity caused by the use of other PGRs have also been described in previous studies [10,11].

Salicylic acid (SA), an endogenous plant defense hormone, also plays a pivotal role in the development of systemic acquired resistance (SAR) in plants [12]. Hence, the field, which focuses on the significant protection provided to diverse plants (especially food crops) against various surrounding-growth stresses by SA application to increase grain productivity and food quality, has been of interest to many researchers. For example, the findings reported by El Sherbiny et al. showed that exogenous 700 µM SA application (sprayed at 15, 30, and 45 days after rice seedling transplantation) had the greatest effect on relieving the negative influence induced by water deficiency and produced the highest values of grain yield [13]. Moreover, Wang et al. reported that SA promoted yield productivity in winter wheat when it was sprayed with 100 µM SA at the jointing stage by increasing resistance to the low-temperature stress that frequently occurs in late spring [14]. Similarly, Tahjib-Ul-Arif et al. reported increases in photosynthesis, antioxidant ability and yield in maize treated with a 1 mM SA solution under salinity stress [15]. Furthermore, the positive roles of SA application in mitigating or resisting biotic and abiotic stresses in various kinds of plants have been well illustrated and summarized in recent literature reviews [12,16,17].

Despite the number of studies on SA and its functions, few reports have been published on the regulatory effects of the exogenous application of SA at the heading stage on aroma accumulation in fragrant rice. Hence, a pot experiment employing three different levels of salicylic acid at the heading stage was conducted in this study with the aims of (1) investigating the impacts of exogenous SA application on aroma and yield to increase the economic quality of fragrant rice. (2) The influence of SA application on the 2-AP synthesis mechanism was researched by studying the

related indices of 2-AP synthesis. (3) The effects of SA on the resistance of fragrant rice were studied by assessing the SOD, POD, and CAT activities and the soluble protein, $H_2O_2$ and MDA contents.

## Methods

### Site description and soil characteristics

The experiments were conducted in outdoor potted conditions under a typical subtropical monsoon climate from July to November 2021, aiming to simulate field meteorological environments as closely as possible. The climatic data of the experimental site are shown in Fig 1. The experimental soil was sandy loam, and the detailed characteristics were as follows: 1.91 g·kg⁻¹ total nitrogen, 1.11 g·kg⁻¹ total phosphorous, 21.24 g·kg⁻¹ total potassium, 22.67 g·kg⁻¹ total organic matter, and a pH value of 6.27.

### Experimental materials

The seeds of two fragrant rice cultivars, i.e., *Meixiangzhan* (variety right number: CNA20162427.9) and *Yuxiangyouzhan* (variety right number: CNA20110880.8), which were provided by the Guangdong Academy of Agricultural Science (Guangzhou, Guangdong Province, China), were selected for this study because of their higher market value and better aroma quality. The detailed processes of disinfection and germination of seeds and young seedling cultivation for both cultivars abided by the methods of Zhang et al. [8]. After that, 8 g of compound fertilizer (N-$P_2O_5$-$K_2O$ = 16-16-16, Yara, Norway) was applied as base fertilizer, and the three-leaf-stage seedlings were transplanted into plastic pots (35 cm × 24 cm, filled with 11.5 kg of sun-dried soil) with four seedlings per hill and five hills per pot. During fragrant rice growth, 5 g of the same compound fertilizer (N-$P_2O_5$-$K_2O$ = 16-16-16) was added as topdressing at the tilling stage. Fertilization practices were consistent across all experimental treatments to ensure uniform nutrient supply. While compound fertilizer is essential for rice growth and yield, its application was standardized across all pots, minimizing potential confounding effects on the results of salicylic acid (SA) treatments. Moreover, other farming practices, including water management, disease prevention, injurious insect control, and weeding, were consistent with local agricultural practices.

### Experimental design

At the initial heading stage, a completely randomized pot experiment was arranged with Salicylic acid (SA, Macklin, China, Cat. No. S817529, reagent grade, 99.5% purity) at four treatments (consisting of 0, 0.1, 0.5, and 1 mM SA, referred to as

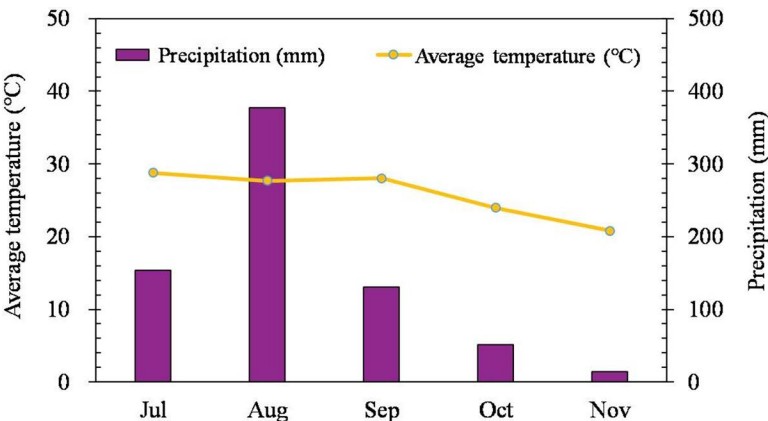

**Fig 1. Average temperature and precipitation data from July and November in the experimental site.**

CK, SA1, SA5, and SA10, respectively). For each variety, each treatment contained 18 pots to ensure sufficient rice plants for sampling during subsequent treatment periods. The determination of SA concentration in each treatment was based on previously published literature by Mohammed et al. and Yang et al. [18,19].

### Sampling and measurement

The experimental treatments (SA spraying) were applied at the initial heading stage (October 5th, 2021). Sampling was conducted at 1 day, 10 days, and 20 days after spraying (AS), and the mature stage (MS, approximately 30 days after the intial sampling). At each experimental stage, the leaves, stems and grains of three representative fragrant rice plants were separated quickly with scissors, soaked immediately in liquid nitrogen for 15 s and stored in a −80°C freezer to stop all biological reactions inside and outside the cell and maintain enzyme activities and substance contents to the maximum extent. Moreover, at MS, mature grains were harvested rapidly from four representative fragrant rice plants and stored at −20°C for 2-AP determination.

### Determination of 2-AP content, yield and yield-related traits

The determination of 2-AP content in grains via MS was performed according to the methods of Ren et al. [2] The fresh and mature grains were rapidly ground to powder manually under freezing conditions in an appropriate amount of liquid nitrogen, and approximately 2 g of ground powder was mixed with 10 mL of dichloromethane for 2-AP extraction through ultrasonication for 4 h. The supernatant of the 2-AP extraction solution was subsequently measured via a GCMS-QP 2010 Plus instrument (produced by Shimadzu Corporation, Japan). The 2-AP content was expressed as $\mu g \cdot g^{-1}$ fresh weight (FW).

Moreover, on the basis of the methods of Zhang et al. [8], productive tillers were calculated from thirty randomly chosen representative rice plants, and the seed-setting rate (%), 1000-grain weight (g) and yield per pot ($g \cdot pot^{-1}$) were evaluated when the harvested grains from five randomly selected pots reached constant weight under continuous sun-drying (which means that the moisture content of the grains decreased by approximately 14%).

### Determination of the contents of substances and enzyme activities related to 2-AP biosynthesis

The substance contents (P5C, proline, and pyrroline) and PDH (EC: 1.5.99.8) activities were detected via the method of Ren et al. [2]. Absorbance measurements were performed using a microplate reader (Biotek, Epoch, USA) at 440 nm, 520 nm, and 430 nm, respectively. For quantification, the molar absorption coefficients ($\varepsilon$) were assumed as 2.58 mM$^{-1}$·cm$^{-1}$ (440 nm) and 1860 cm$^{-1}$ (430 nm). All measurements were conducted in four replicates, and the contents of P5C, pyrroline, and proline were expressed in $\mu mol \cdot g^{-1}$ FW, $\mu g \cdot g^{-1}$ FW, and $mmol \cdot g^{-1}$ FW, respectively. The reliability of absorbance readings was ensured by using consistent instrument calibration, replicate measurements, and validated protocols aligned with prior literature.

### Determination of physiological indices related to antioxidant attributes

As described by Zhang et al. [8], the absorbances of $H_2O_2$ and soluble protein were measured at 410 nm (the maximum absorption wavelength of the $H_2O_2$-colorimetric reagent complex) and 595 nm (using the Coomassie Brilliant Blue method, where the absorbance at 595 nm is proportional to protein concentration within the range of 0–100 μg/ml). For MDA (malondialdehyde) measurement, absorbances were recorded at 470, 649, and 665 nm (the dual-wavelength method at 649 nm and 665 nm is used to calculate MDA content by subtracting nonspecific absorption at 470 nm, while 470 nm serves as a reference to correct for background interference). The units for $H_2O_2$, soluble protein, and MDA were expressed as $\mu g \cdot g^{-1}$·FW, $\mu g \cdot g^{-1}$·FW, and $\mu mol \cdot g^{-1}$·FW, respectively.

Moreover, according to the methods of Zhang et al. [8], SOD (EC: 1.15.1.1) activity was measured at 560 nm using the nitrogen blue tetrazolium (NBT) photochemical reduction method. The assay detects the inhibition of NBT reduction

 

by superoxide anions, with absorbance at 560 nm inversely proportional to SOD activity. POD (EC: 1.11.1.7) activity was determined at 470 nm via the guaiacol peroxidase method. The increase in absorbance at 470 nm reflects the oxidation of guaiacol to tetraguaiacol by hydrogen peroxide, catalyzed by POD. CAT (EC: 1.11.1.6) activity was assayed at 240 nm by monitoring the decrease in absorbance of $H_2O_2$ at this wavelength. The rate of $H_2O_2$ decomposition (absorbance decline at 240 nm) is proportional to CAT activity. All enzyme activities were expressed as $U \cdot g^{-1}$ FW (SOD), $U \cdot g^{-1} \cdot min^{-1}$ FW (POD), and $U \cdot g^{-1} \cdot min^{-1}$ FW (CAT), respectively, with measurements performed in four plicate biological replicates.

### Statistical analysis

Microsoft Excel 2010 (Microsoft Corporation, New Mexico, USA) and Origin 9.0 (OriginLab Corporation, Northampton, Massachusetts, USA) were used for data curation and graph drawing. All measurements were performed with four biological replicates. Before statistical analysis, data were tested for normality (Shapiro-Wilk test) and homogeneity of variances (Levene's test). For normally distributed data with homogeneous variances, one-way analysis of variance (ANOVA) was performed using SPSS Statistics 19.0 (Analytical, Armonk, New York, USA). The significant differences among means were separated via the least significant difference (LSD) test at a probability level of 5% ($P < 0.05$). Moreover, the correlations between the detected parameters and principal component analysis (PCA) results were obtained via MetaboAnalyst software (http://www.metaboanalyst.ca).

## Results

### Grain yield

Salicylic acid (SA) treatment significantly enhanced the seed-setting rate and yield, while showing no significant differences in the number of panicles per hill, number of grains per panicle, or 1000-grain weight across treatments (Table 1). In *Meixiangzhan* and *Yuxiangyouzhan*, SA treatment increased the seed-setting rate by 8.9% and 5.5%, respectively, and yield by 12.0% and 8.1%, respectively, compared to the control. The SA5 treatment consistently achieved the highest seed-setting rate and yield in both cultivars. Notably, *Meixiangzhan* exhibited significantly fewer grains per panicle and lower 1000-grain weight than *Yuxiangyouzhan*, though it had a higher number of panicles per hill.

**Table 1. Effects of salicylic acid application on grain yield and yield-related traits in fragrant rice.**

| Cultivar | Treatment | Panicle number per hill | Grains per panicle | Seed-setting rate (%) | 1000-grain weight (g) | Yield (g·pot⁻¹) |
|---|---|---|---|---|---|---|
| *Meixiangzhan* | CK | 12.81±0.17a | 110.01±3.29a | 71.69±2.38c | 19.05±0.60a | 77.19±1.45b |
| | SA1 | 12.69±0.11a | 107.74±5.63a | 76.29±0.58bc | 19.03±0.01a | 86.32±1.46a |
| | SA5 | 12.72±0.06a | 108.22±3.41a | 81.34±1.19a | 18.67±1.08a | 88.22±1.22a |
| | SA10 | 12.75±0.09a | 109.00±1.25a | 76.68±1.31ab | 18.55±0.22a | 84.90±4.21a |
| | **Mean** | **12.74A** | **108.74B** | **76.50A** | **18.82B** | **84.16B** |
| *Yuxiangyouzhan* | CK | 11.75±0.05a | 125.15±1.71a | 78.40±0.86b | 21.25±0.52a | 101.90±2.84b |
| | SA1 | 11.88±0.13a | 123.03±6.22a | 82.23±0.95a | 21.75±0.31a | 106.25±5.70ab |
| | SA5 | 11.81±0.11a | 125.49±0.99a | 83.97±0.76a | 21.72±0.12a | 116.02±1.54a |
| | SA10 | 11.69±0.08a | 126.88±4.74a | 82.01±0.85a | 21.61±0.48a | 108.05±2.96ab |
| | **Mean** | **11.78B** | **125.14A** | **81.65A** | **21.58A** | **108.05A** |

Various letters indicate significant differences according to the LSD test at *P<0.05.* Capital letters indicate comparisons between cultivars; lowercase letters indicate comparisons between treatments within a cultivar.

 

## 2-AP content

The 2-AP content exhibited a significant increase with increasing SA treatment concentrations (Fig 2). In *Meixiangzhan*, the 2-AP content under SA1, SA5, and SA10 treatments was 20.7%, 68.2%, and 10.3% higher, respectively, compared to the CK treatment. For *Yuxiangyouzhan*, the 2-AP content in SA1, SA5, and SA10 treatments was 6.0%, 22.7%, and 7.9% higher, respectively, than in the CK treatment. Notably, the SA5 treatment consistently yielded the highest 2-AP content across both cultivars, highlighting its optimal role in enhancing aromatic compound accumulation.

## Activities of PDH and contents of proline, P5C and pyrroline

For the two cultivars, salicylic acid (SA) treatment generally increased PDH activity across all sampling stages, with the exception that in *Meixiangzhan* leaves at 1 day after salicylic acid treatment (1 d AS), SA1 treatment did not lead to a significant decrease (Fig 3). Specifically, in *Meixiangzhan*, SA1 treatment resulted in the highest PDH activity in grains at 1 d AS and in stems at maturity stage (MS). SA5 treatment exhibited the highest PDH activity in grains at 10 days after salicylic acid treatment (10 d AS), 20 days after salicylic acid treatment (20 d AS), and MS, as well as in leaves at 1 d AS and 10 d AS. Moreover, SA10 treatment had the highest PDH activity values in leaves at 20 d AS and MS, and in stems at 1 d AS, 10 d AS, and 20 d AS.

Regarding the contents of substances related to 2-AP synthesis, SA5 treatment led to the highest proline content in grains at 1 d AS, with an average increase of 58.8% compared to the CK treatment (Fig 4A). For *Meixiangzhan*, SA1 treatment resulted in the maximum proline content at later sampling stages (10 d AS, 20 d AS, and MS). In stems, SA5 treatment led to the highest proline levels at 20 d AS and MS (Fig 4C). In *Yuxiangyouzhan* at MS, SA5 treatment resulted in the highest P5C content (Fig 5). Across different sampling stages and treatments, both cultivars showed varying degrees of P5C content changes under salicylic acid treatments. In some cases, there were significant increases compared to the CK treatment, while in others, the differences were not significant. For example, in *Meixiangzhan* at certain stages, SA1 and SA5 treatments both led to elevated P5C levels, but the magnitudes of increase and significance varied. In both cultivars, SA10 treatment significantly decreased the pyrroline content in grains at MS. Across all salicylic acid concentration treatments, both cultivars generally showed a significant increase or no significant difference in pyrroline content in other tissues and sampling stages (Fig 6). In general, salicylic acid treatments had diverse effects on the contents of proline, P5C, and pyrroline, with SA5 often promoting higher contents in specific tissues and sampling stages.

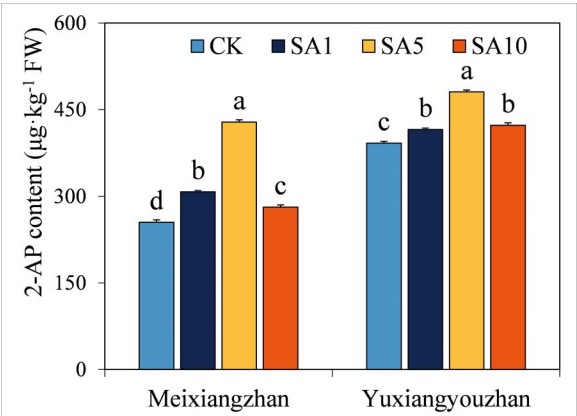

**Fig 2. Effects of salicylic acid application on grain 2-AP content at MS in fragrant rice.** A lowercase letter denotes statistically significant differences between treatments within a cultivar, as determined by the LSD test at a significance level of *P<0.05*. Bars are SE (n=4).

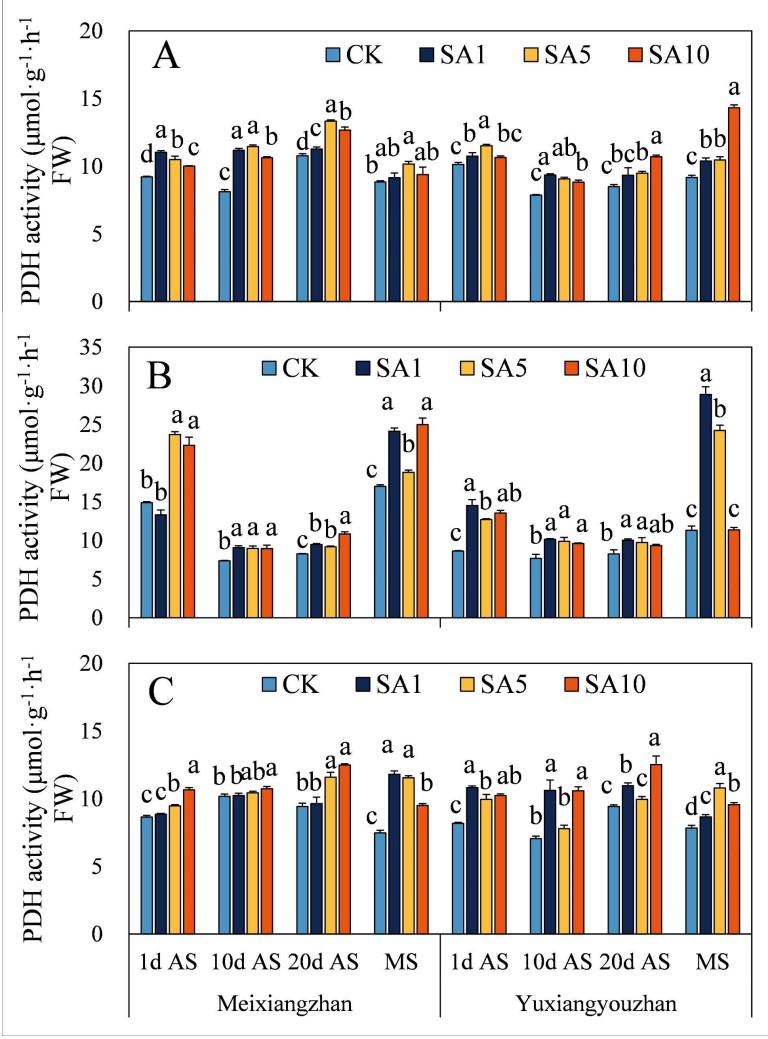

**Fig 3. Effects of salicylic acid application on PDH activities in grains (A), leaves (B) and stems (C) for fragrant rice.** A lowercase letter denotes statistically significant differences between treatments within a cultivar, as determined by the LSD test at a significance level of *P<0.05*. Bars are SE (n=4).

## Antioxidant enzyme activity

The SOD activity of the two fragrant rice cultivars increased in response to SA treatment, and the activity of both cultivars reached a significant level at 1 d AS, except for the significant decrease in SOD activity in the stems at MS and in the grains at 20 d AS under the SA5 and SA10 treatments for *Meixiangzhan* (Fig 7). For both fragrant rice cultivars, POD activity significantly increased, but there was no significant difference (Fig 8). There was a significant decrease in the CAT activity of fragrant rice at 1 d AS, but the level of CAT activity increased at 10 d, 20 d AS and MS and reached a significant level at 10 d AS and MS (Fig 9).

## Contents of MDA, H₂O₂ and soluble protein

At 1 d AS, the MDA content significantly increased in both fragrant rice cultivars, except for *Yuxiangyouzhan*, whose content did not differ between the two rice cultivars under SA1 treatment (Fig 10). However, at 10 d, 20 d AS and

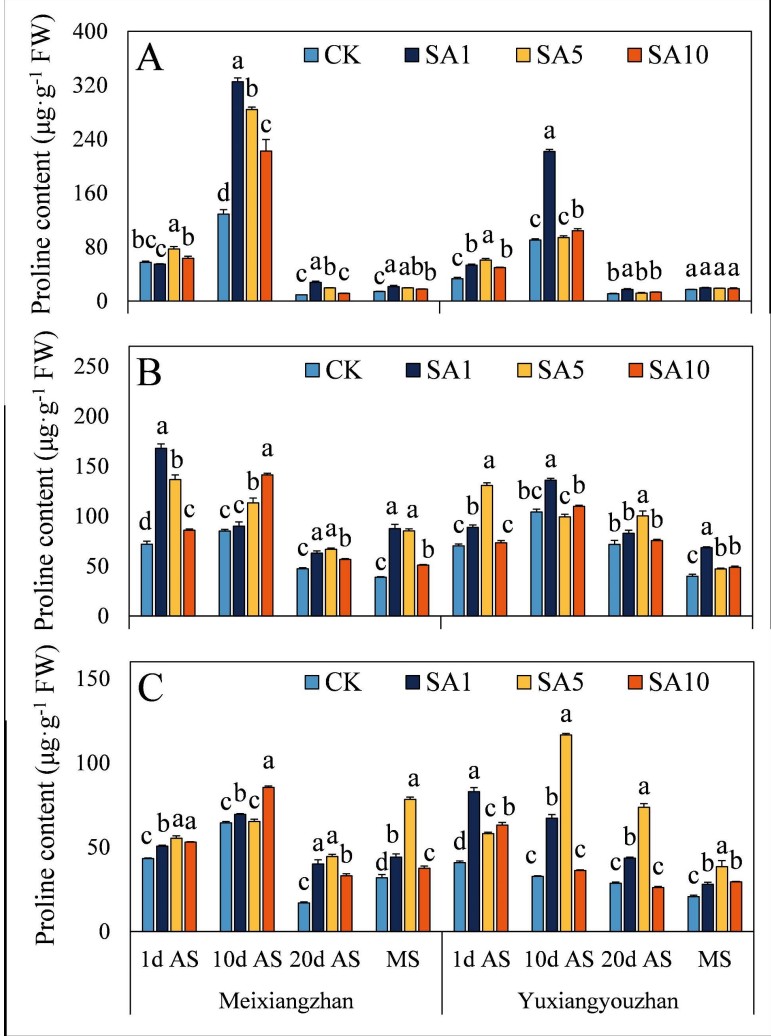

**Fig 4. Effects of salicylic acid application on proline content in grains (A), leaves (B) and stems (C) for fragrant rice.** A lowercase letter denotes statistically significant differences between treatments within a cultivar, as determined by the LSD test at a significance level of *P<0.05*. Bars are SE (n=4).

MS, the MDA content significantly decreased, except at 20 d AS, in the grains and stems of *Yuxiangyouzhan*. Additionally, the SA5 treatment resulted in the lowest MDA content at 20 d AS and MS in the grains and at 20 d AS in the stems of *Yuxiangyouzhan* and at 10 d AS and MS in the leaves of *Meixiangzhan*. The $H_2O_2$ content significantly increased in both cultivars following SA treatment, and the SA1 treatment resulted in the highest $H_2O_2$ content for *Yuxiangyouzhan* in all tissues and for *Meixiangzhan* in the grains (Fig 11). At 10 d and 20 d AS and MS, the $H_2O_2$ content significantly decreased in the grains and stems but increased in the leaves. Compared with that in the CK treatment, the soluble protein content in the SA treatment significantly increased or did not differ between the two cultivars, except in the grains of *Meixiangzhan* at 20 d AS under the SA10 treatment (Fig 12). Notably, the grain soluble protein contents of the two cultivars were recorded in the SA5 and SA1 treatments at 20 d AS and MS, respectively (Fig 12A).

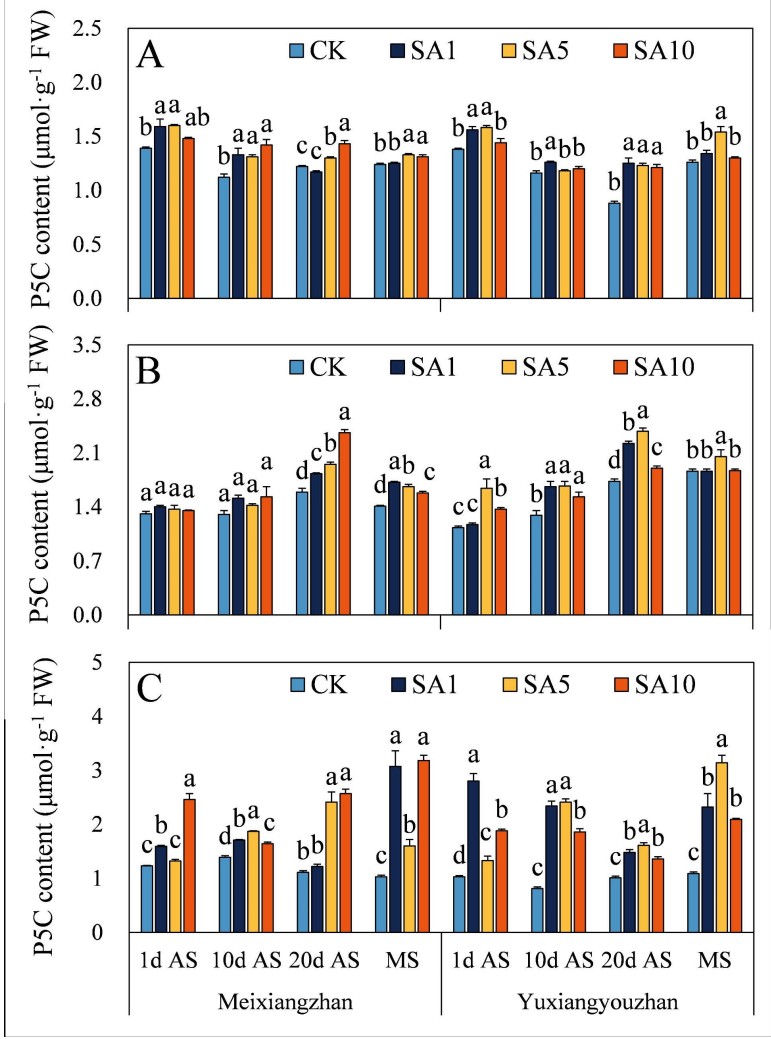

**Fig 5. Effects of salicylic acid application on P5C content in grains (A), leaves (B) and stems (C) for fragrant rice.** A lowercase letter denotes statistically significant differences between treatments within a cultivar, as determined by the LSD test at a significance level of *P<0.05*. Bars are SE (n=4).

## Multiple comparative analyses of 2-AP content, yield and related parameters

On the basis of all the measured parameters, the 25 parameters that were highly correlated with the grain 2-AP content and yield were determined in fragrant rice when foliar SA was applied (Fig 13A and B). For the 2-AP content, the 5 closely positively correlated parameters were the leaf proline content at 20 d AS (Pro_L_20d AS), the leaf P5C content at MS (P5C_L_MS), the pyrroline content in stems and grains at 1 d AS (Pyr_S_1d AS and Pyr_G_1d AS), and the PDH activity at 1 d AS (PDH_G_1d AS) (Fig 13 A). For yield, stem SOD activity at MS (SOD_S_MS), 1000-grain weight (GW), the seed-setting rate (SR), the leaf MDA content at 1 d AS (MDA_L_1d AS), and the grain $H_2O_2$ content at 10 d AS (H2O2_G_10d AS) were the 5 closely positively correlated parameters. Additionally, as a result of the PCA for 2-AP, two principal components accounted for greater than 90% of the variance in the parameters related to 2-AP biosynthesis (Fig 13 C). The treatment groups differed significantly, while the high-dose group (SA10) was close to the control and overlapped partially. Moreover, the SA5 treatment group showed a significant difference and high score plot in PC2,

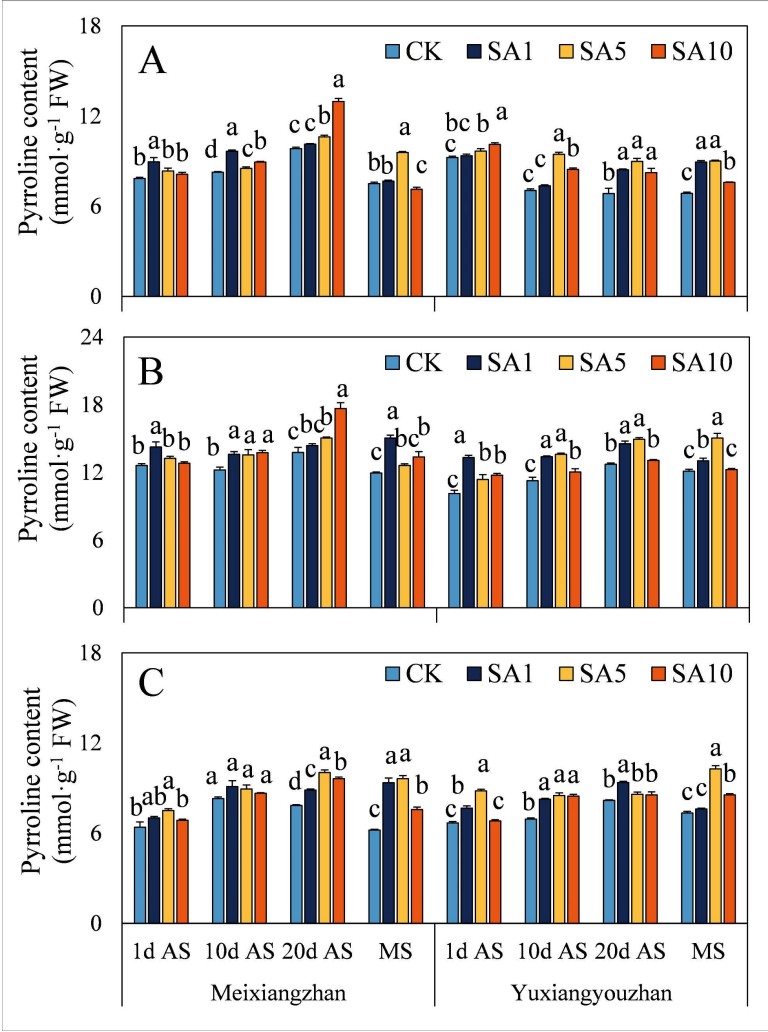

**Fig 6. Effects of salicylic acid application on pyrroline content in grains (A), leaves (B) and stems (C) for fragrant rice.** A lowercase letter denotes statistically significant differences between treatments within a cultivar, as determined by the LSD test at a significance level of *P<0.05*. Bars are SE (n=4).

indicating the best efficacy for 2-AP accumulation in response to 0.5 mM SA application. However, two principal components accounted for more than 67% of the variance in the parameters related to yield formation and antioxidant properties according to PCA (Fig 13 D). There was no overlap between each treatment group and the control group and a significant difference, but there was no significant difference between each treatment component with interlaps. This suggests that foliar SA application has better efficacy for yield formation, whereas the concentration applied was not different in this study.

## Discussion

With the average growth rate of the world population reaching approximately 1.10%, the rapidly increasing global population will reach 9.5 billion by 2050; thus, increasing grain production from limited arable farmland while protecting soil fertility and productivity to establish a strategy for sustainable agricultural development has become a vital problem that

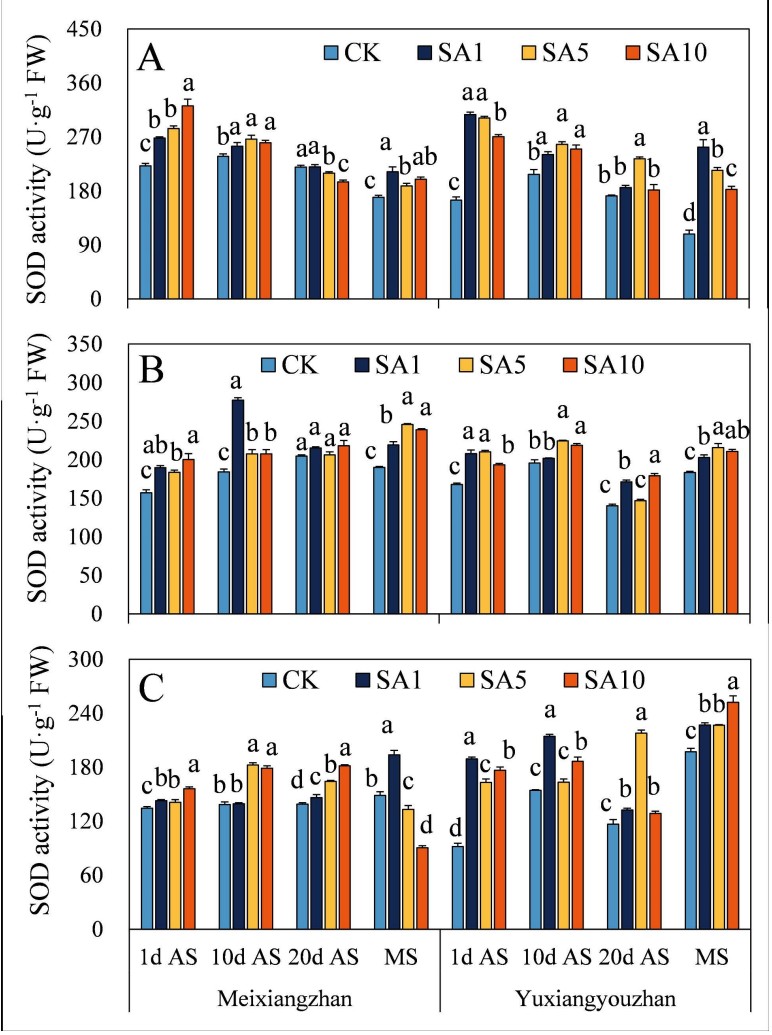

**Fig 7. Effects of salicylic acid application on SOD activities in grains (A), leaves (B) and stems (C) for fragrant rice.** A lowercase letter denotes statistically significant differences between treatments within a cultivar, as determined by the LSD test at a significance level of *P < 0.05*. Bars are SE (n = 4).

desperately needs to be solved [20]. Salicylic acid (SA), celebrated for its low cost, biodegradability, and high efficiency, has emerged in numerous studies as a viable tool to boost crop productivity under both normal and stressful conditions [21]. For instance, Yang et al. demonstrated that foliar application of 0.5 mmol·L⁻¹ SA significantly increased rice yields across five different sowing dates [22]. In our study, exogenous SA application at the heading stage enhanced average yields by 12.04% in *Meixiangzhan* and 8.06% in *Yuxiangyouzhan*, with the seed-setting rate being a primary driver of this increase (Table 1). This outcome is likely attributed to SA's role in promoting spikelet development, maintaining grain fertility, accelerating grain filling, and optimizing resource allocation during reproductive growth [9,18]. These findings align with Yang et al., who observed that 0.5 mM SA spray at the heading-flowering stage significantly improved grains per panicle and seed-setting rate, ultimately enhancing yield [19]. Collectively, these results underscore SA's potential to synchronize physiological processes and maximize yield components in fragrant rice, offering a sustainable approach to meet rising food security demands.

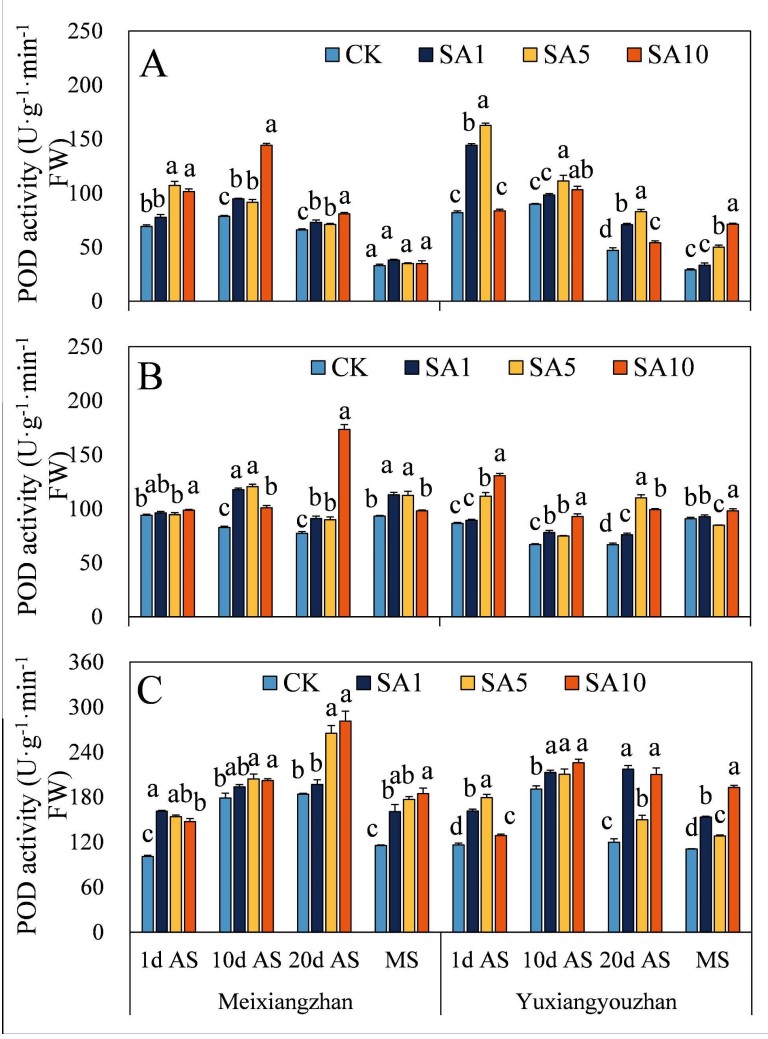

**Fig 8. Effects of salicylic acid application on POD activities in grains (A), leaves (B) and stems (C) for fragrant rice.** A lowercase letter denotes statistically significant differences between treatments within a cultivar, as determined by the LSD test at a significance level of *P<0.05*. Bars are SE (n = 4).

As the major contributor to grain aroma, the content of 2-AP strongly influences the fragrance, taste, and even economic characteristics of fragrant rice [3]. Consequently, the effects of plant growth regulators on 2-AP accumulation have garnered significant attention in recent years. In this pot experiment, all exogenous SA treatments significantly enhanced grain 2-AP levels in both cultivars: *Meixiangzhan* showed increases of 20.7%, 68.2%, and 10.3% under SA1, SA5, and SA10, respectively, while *Yuxiangyouzhan* exhibited 6.0%, 22.7%, and 7.9% increases (Fig 2). Notably, these findings align with prior research on aromatic rice: Zhang et al. reported that foliar methyl jasmonate (MeJA, 1–2 μM) increased 2-AP by 18.3–61.3% across cultivars [8]. Xing et al. demonstrated that trans-zeatin (20–40 μmol·L⁻¹) boosted 2-AP in Meixiangzhan by 36.6–59.1% [7]. The striking 68.2% increase in 2-AP under SA5 for Meixiangzhan—comparable to the most effective growth regulators in literature—highlights SA's potential as a low-cost, high-efficiency tool for enhancing aromatic quality. These results establish a clear mechanistic link between SA application and 2-AP biosynthesis, reinforcing its utility as a sustainable strategy for aromatic rice improvement.

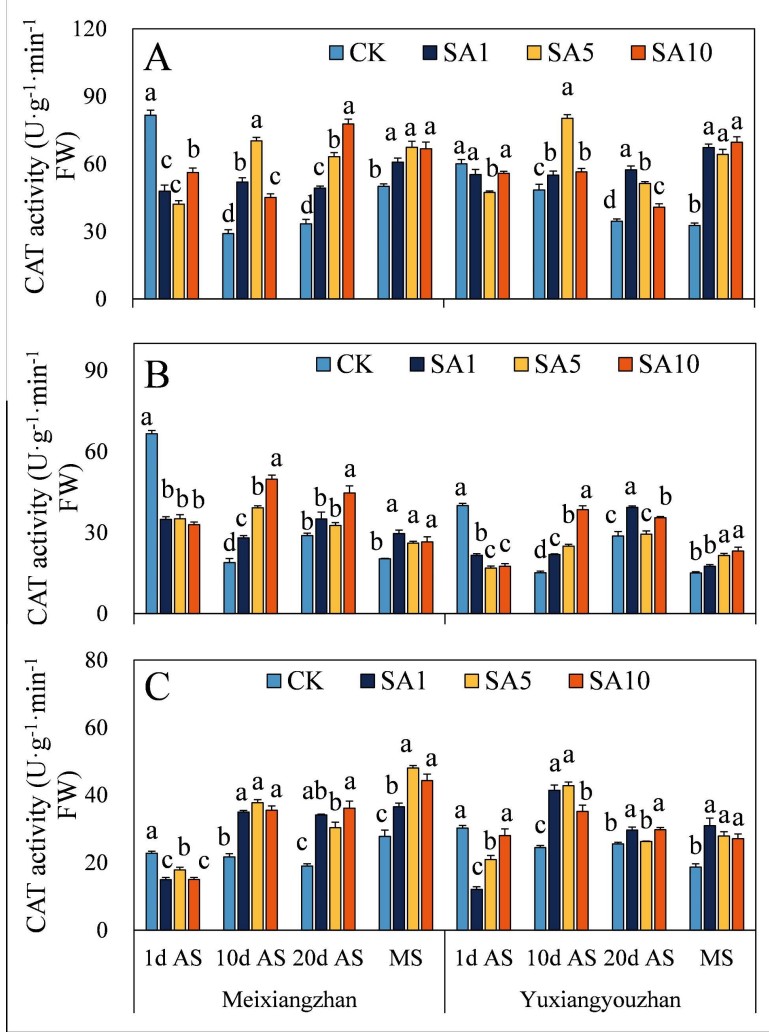

**Fig 9. Effects of salicylic acid application on CAT activities in grains (A), leaves (B) and stems (C) for fragrant rice.** A lowercase letter denotes statistically significant differences between treatments within a cultivar, as determined by the LSD test at a significance level of *P<0.05*. Bars are SE (n=4).

To date, the complete biosynthetic pathway of 2-AP remains incompletely elucidated, though the proline-dependent pathway has been well-documented in published literature. Briefly, proline is converted to P5C via PDH, after which P5C can generate 2-AP through two routes: (1) direct enzymatic or nonenzymatic conversion, or (2) decarboxylation to pyrroline by pyrrole-5-carboxylic acid decarboxylase, followed by further conversion to 2-AP [2]. In this study, foliar SA application enhanced 2-AP content, which correlated with increased levels of proline (except *Meixiangzhan* grains at 1 d AS under SA1, *Yuxiangyouzhan* leaves at 10 d AS under SA5, and stems at 20 d AS under SA10), P5C (except *Meixiangzhan* grains at 20 d AS under SA1), and pyrroline (except *Meixiangzhan* grains at MS under SA10), as well as elevated PDH activity (except *Meixiangzhan* leaves at 1 d AS under SA1). These findings align with prior research by Cheng et al. and Zhang et al., who demonstrated that exogenous methylglyoxal and methyl jasmonate (MeJA) upregulated 2-AP biosynthesis-related physiological indices, ultimately increasing grain 2-AP content at maturity [8,23]. Notably, correlation analysis further revealed strong positive associations between 2-AP content and leaf proline/P5C levels, stem/

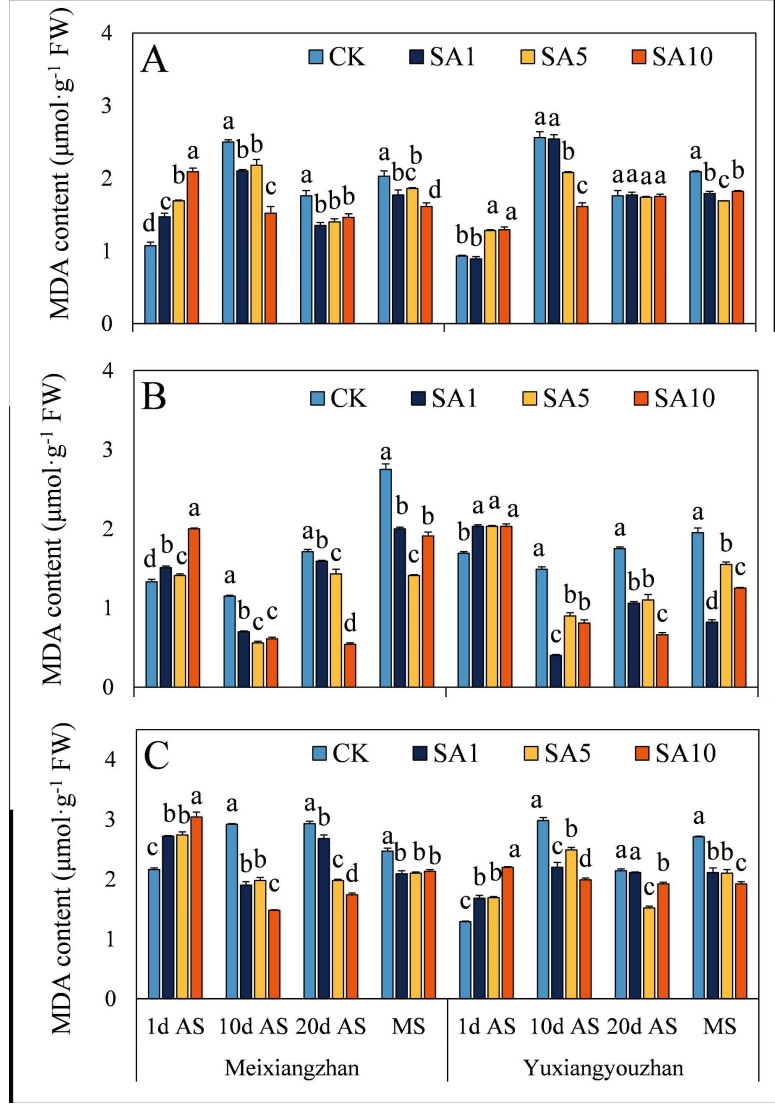

**Fig 10. Effects of salicylic acid application on MDA contents in grains (A), leaves (B) and stems (C) for fragrant rice.** A lowercase letter denotes statistically significant differences between treatments within a cultivar, as determined by the LSD test at a significance level of *P<0.05*. Bars are SE (n=4).

grain pyrroline levels, and PDH activity (Fig 13), underscoring the central role of these metabolites and enzyme activity in SA-mediated 2-AP accumulation. The observed improvements may also be linked to SA-induced enhancements in plant antioxidant capacity, which likely protects biosynthetic enzymes from oxidative damage and sustains metabolic flux toward 2-AP production.

The evolutionary history of plants can also be considered a process in which plants adapt to many changing environmental conditions, defend against different pathogens and resist pest feeding. However, alongside the progress of plant growth and development, plants are forced to endure diverse biotic and abiotic stresses because of their immovable features. The critical functions of plant hormones (particularly abscisic acid, ethylene, jasmonic acid, and salicylic acid) have been shown to mediate and alleviate plant stress responses [24,25]. Since Ross first identified systemic acquired

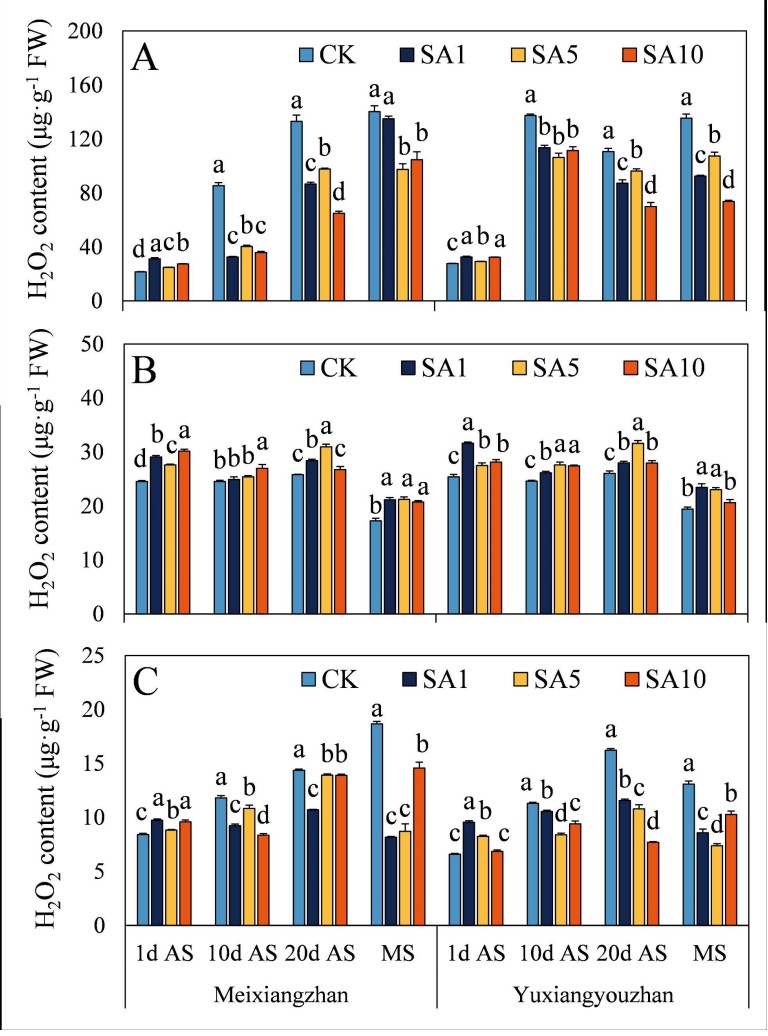

**Fig 11. Effects of salicylic acid application on H$_2$O$_2$ contents in grains (A), leaves (B) and stems (C) for fragrant rice.** A lowercase letter denotes statistically significant differences between treatments within a cultivar, as determined by the LSD test at a significance level of *P<0.05*. Bars are SE (n=4).

resistance (SAR) in 1961, SA has been extensively studied for its role in priming SAR and modulating antioxidant defences via oxidative stress induction in various plant species [21,26,27]. In this study, we investigated how exogenous SA application influences the oxidative resistance of fragrant rice at multiple growth stages. (1) Early-stage response, at 1 day after SA spraying, most experimental tissues exhibited increased MDA (except *Yuxiangyouzhan* grains under SA1), H$_2$O$_2$, and soluble protein contents, as well as elevated SOD and POD activities, while CAT activity decreased (Figs 7–12). This paradoxical pattern can be explained by SA's dual role in stress signaling: firstly, CAT inhibition: as a key enzyme in H$_2$O$_2$ scavenging and a primary component of SA signaling, CAT is inactivated upon SA binding, leading to reduced H$_2$O$_2$ clearance and subsequent ROS accumulation [21,28]. Secondly, oxidative burst and compensatory mechanisms: the resulting H$_2$O$_2$ and ROS surge triggers membrane lipid peroxidation (elevated MDA) but also activates upstream defenses. Increased SOD/POD activities and soluble protein content reflect the plant's attempt to scavenge excess ROS, maintain membrane integrity, and restore cellular homeostasis [26,29]. (2) Subsequent-stage adaptation, in

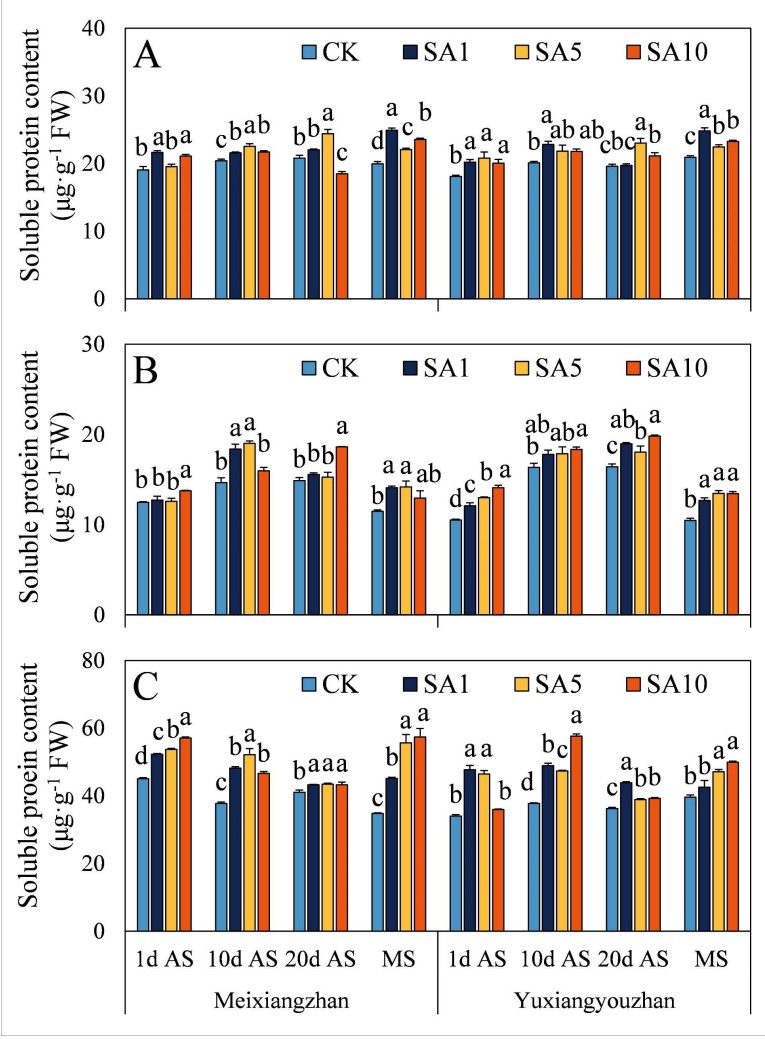

**Fig 12. Effects of salicylic acid application on soluble protein contents in grains (A), leaves (B) and stems (C) for fragrant rice.** A lowercase letter denotes statistically significant differences between treatments within a cultivar, as determined by the LSD test at a significance level of *P<0.05*. Bars are SE (n=4).

later stages, most SA-treated tissues showed decreased $H_2O_2$ and MDA levels (except *Yuxiangyouzhan* grains at 20 d AS under SA1) and increased CAT activity, alongside variable trends in SOD/POD and soluble proteins (e.g., *Meixiangzhan* grains/stems at 20 d AS under SA5/SA10). This shift is likely driven by ROS as secondary messengers: the early-stage oxidative burst primes defense-related gene expression (e.g., SAR markers), enhancing the plant's capacity to neutralize ROS and mitigate lipid peroxidation [30,31]. Sustained SA presence activates long-term stress responses, balancing ROS production and scavenging to avoid self-toxicity [21]. These findings align with SA's known role in inducing "stress priming": initial oxidative stress triggers adaptive responses that ultimately enhance stress resilience. The transient CAT inhibition and ROS surge at 1 d AS represent a critical early step in SA-mediated defense activation, while the subsequent recovery of antioxidant systems highlights the plant's ability to optimize resource allocation for both stress tolerance and growth [27,31].

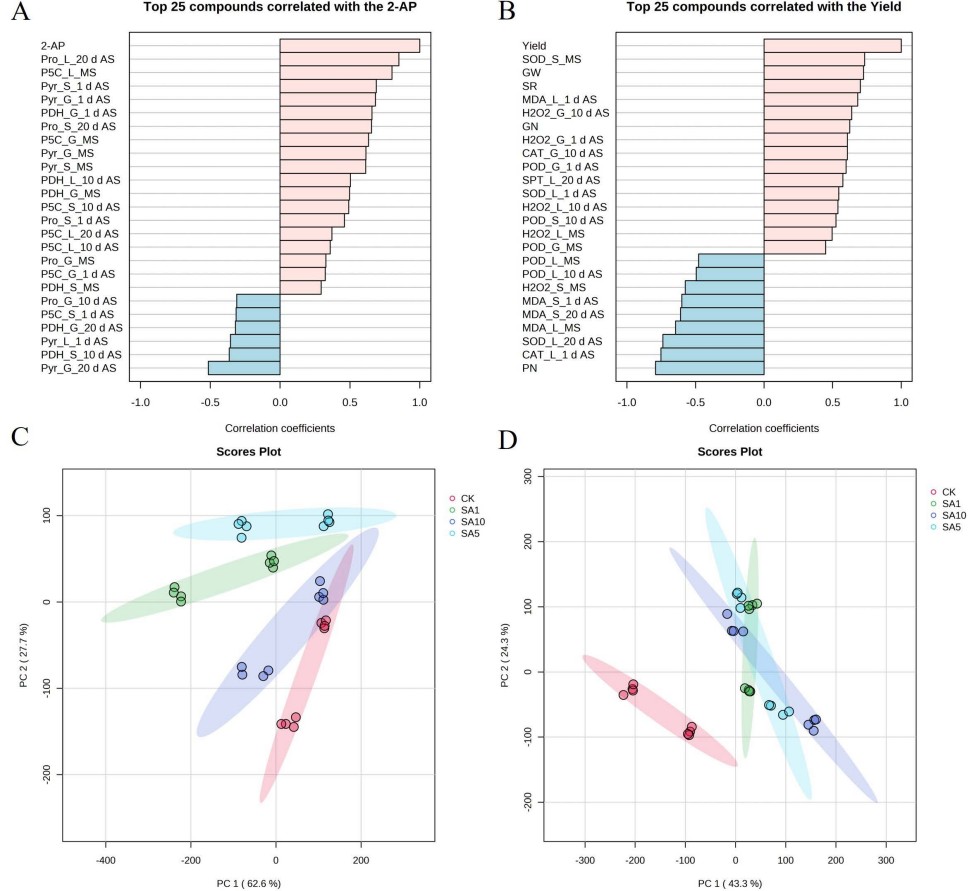

**Fig 13. The top 25 parameters correlated with the grain 2-AP (A) and yield (B), score plots with 2 dimensions for 2-AP (C) and yield (D) by PCA.** 2-AP, 2-acetyl-1-pyrroline content; Pro, proline; P5C, pyrroline-5-carboxylic acid; Pry, pyrroline; PDH, proline dehydrogenase; SOD, superoxide dismutase; POD: peroxidase; CAT: catalase; MDA: malondialdehyde; SPT: soluble protein; AS, after spraying; MS, maturity stage; G, grain; L, leaf; S, stem. GW, 1000-grain weight; SR, seed-setting rate; PN, panicle number per hill; GN, grains number per panicle.

## Conclusions

Salicylic acid treatment significantly increased the 2-AP content in the grains of the two cultivars at the maturity stage ($P < 0.05$) and promoted yield performance. Among the treatments, SA5 (5 mM) showed the highest comprehensive performance: while its yield was not significantly different from SA10 (10 mM) ($P > 0.05$), SA5 exhibited the maximum yield trend and had significantly higher 2-AP content than all other treatments ($P < 0.05$). Additionally, SA5 treatment enhanced the activities of enzymes (PDH) and precursors (proline, P5C, and pyrroline) involved in 2-AP synthesis, as well as the antioxidant properties of fragrant rice plants. The activities of antioxidant enzymes (SOD, POD, and CAT) under SA5 treatment significantly influenced physio-biochemical attributes, further regulating 2-AP formation and yield-related traits. Although yield did not differ significantly between SA5 and SA10, the superior 2-AP content in SA5 (a key aromatic compound) and its stable performance in other physiological indices (e.g., proline accumulation, antioxidant enzyme activities) were critical factors for selecting it as the optimal treatment. The results indicate that SA application can regulate 2-AP biosynthesis and yield through both direct and indirect pathways, with proline content being a major influencing index under salicylic acid treatment. However, the detailed molecular mechanisms require further exploration. This study provides a feasible approach for using phytohormones to enhance aromatic rice production.

## Acknowledgments

We extend our sincere gratitude to the Guangdong Academy of Agricultural Sciences for generously providing the rice seeds (*Meixiangzhan* and *Yuxiangyouzhan*) used in this study. We also thank the teachers from the Rice Research Laboratory of South China Agricultural University for their valuable guidance and support throughout the research process. Additionally, we appreciate all members of our research team for their dedicated assistance in fieldwork and laboratory analyses.

## Author contributions

**Conceptualization:** Siren Cheng, Yong Ren.

**Data curation:** Siren Cheng, Xueqing Wang, Likai Zheng, Jinyue Liao, Qingqing Li.

**Formal analysis:** Siren Cheng, Yong Ren.

**Funding acquisition:** Siren Cheng.

**Investigation:** Xueqing Wang, Likai Zheng, Jinyue Liao, Qingqing Li.

**Methodology:** Yong Ren.

**Visualization:** Siren Cheng.

**Writing – original draft:** Siren Cheng.

**Writing – review & editing:** Siren Cheng, Yong Ren.

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
