## [Decision Letter · Decision Letter 0]

Dear Dr. Ren,

Thank you for submitting your manuscript to PLOS ONE. After careful consideration, we feel that it has merit but does not fully meet PLOS ONE’s publication criteria as it currently stands. Therefore, we invite you to submit a revised version of the manuscript that addresses the points raised during the review process.

We look forward to receiving your revised manuscript.

Kind regards,

Trung Quang Nguyen

Academic Editor

PLOS ONE

Journal Requirements:

This research was supported by the Natural Science Foundation of Guangxi Zhuang Autonomous Region (grant number: 2021GXNSFBA196084 and 2023GXNSFBA026307), the Middle-aged and Young Teachers' Basic Ability Promotion Project of Guangxi (grant number: 2023KY0609).

This research was supported by the Natural Science Foundation of Guangxi Zhuang Autonomous Region (grant number: 2021GXNSFBA196084 (S.C.)and 2023GXNSFBA026307(Y.R.)), the Middle-aged and Young Teachers' Basic Ability Promotion Project of Guangxi (grant number: 2023KY0609(Y.R.)).

4. Please remove your figures from within your manuscript file, leaving only the individual TIFF/EPS image files, uploaded separately. These will be automatically included in the reviewers’ PDF**.**

**5.** Please include captions for your Supporting Information files at the end of your manuscript, and update any in-text citations to match accordingly. Please see our Supporting Information guidelines for more information: http://journals.plos.org/plosone/s/supporting-information.

Reviewers' comments:

Reviewer's Responses to Questions

**Comments to the Author**

1. Is the manuscript technically sound, and do the data support the conclusions?

Reviewer #1: Partly

2. Has the statistical analysis been performed appropriately and rigorously?

Reviewer #1: I Don't Know

3. Have the authors made all data underlying the findings in their manuscript fully available?

Reviewer #1: Yes

4. Is the manuscript presented in an intelligible fashion and written in standard English?

Reviewer #1: Yes

Reviewer #1: The authors need to present information on the salicylic acid used in the experiment (information of the product).

The figure 1 in the article shows the average temperature and precipitation data suggesting the experiment was not carried out in the green house, where there reasons the experiments was not conducted in the green house? Also, if the experiments were carried out between July to November where the number of days day1, day 20 to which the experiment was carried out, done in each month between July and November. The error bars on the charts suggest replicates how many replicates was done for each condition?

In the manuscript the authors mentioned using compound fertilizer in page 107 what is the constituents of the compound fertilizer? Was the compound fertilizer added to all pots condition? Could the compound fertilizer have an effect on the yield?

line 145 what instruments was used in reading the absorbance and how reliable was the absorbance reading for quantification.

line 153 why was soluble protein read at 595 nm

Statistical tool was applied on the results but was not used in interpreting the results in situations where there are no significant difference in the outcome of the application of salicylic acid for example in yield the author goes ahead to mention the SA5 condition having the best outcome when there was no significant difference with SA10.

**Do you want your identity to be public for this peer review?** For information about this choice, including consent withdrawal, please see our Privacy Policy

Reviewer #1: No

---

## [Author Response · Author response to Decision Letter 1]

17 May 2025

Extract of referee opinions and editorial suggestions for manuscript ref. No.: PONE-D-25-08260

Editors' comments:

ANS: Thank you for your feedback on our manuscript. We have carefully reviewed the additional requirements and ensured that our revised manuscript adheres to PLOS ONE’s style guidelines.

ANS: Thank you for bringing this to our attention. We confirm that the corresponding author has an ORCID iD and it has been validated in Editorial Manager. The process was completed by accessing ‘Update my Information,’ clicking the Fetch/Validate link next to the ORCID field, and authenticating the iD via the ORCID site. Please let us know if any further verification is needed.

This research was supported by the Natural Science Foundation of Guangxi Zhuang Autonomous Region (grant number: 2021GXNSFBA196084 and 2023GXNSFBA026307), the Middle-aged and Young Teachers' Basic Ability Promotion Project of Guangxi (grant number: 2023KY0609).

This research was supported by the Natural Science Foundation of Guangxi Zhuang Autonomous Region (grant number: 2021GXNSFBA196084 (S.C.)and 2023GXNSFBA026307(Y.R.)), the Middle-aged and Young Teachers' Basic Ability Promotion Project of Guangxi (grant number: 2023KY0609(Y.R.)).

ANS: Thank you for your careful review of our manuscript. We have addressed the funding information as required:

(1) All funding-related text has been deleted from the manuscript’s Acknowledgments section. The revised Acknowledgments now only include non-financial acknowledgments.

(2) We confirm that the current Funding Statement in the online submission form is accurate: " This research was supported by the Natural Science Foundation of Guangxi Zhuang Autonomous Region (grant number: 2021GXNSFBA196084 (S.C.)and 2023GXNSFBA026307(Y.R.)), the Middle-aged and Young Teachers' Basic Ability Promotion Project of Guangxi (grant number: 2023KY0609(Y.R.))."

Please proceed to update the online submission form with the above statement. We appreciate your guidance in ensuring compliance with PLOS ONE’s guidelines.

4. Please remove your figures from within your manuscript file, leaving only the individual TIFF/EPS image files, uploaded separately. These will be automatically included in the reviewers’ PDF.

ANS: Thank you for your guidance. We have addressed the figure formatting requirements as follows: All image files have been deleted from the main manuscript document. Only figure citations and captions remain. The manuscript now contains text only, in compliance with PLOS ONE’s guidelines. And, all figures have been saved as individual TIFF files with the following names: Fig1.tif, Fig2.tif, Fig3.tif, …�Fig13.tif. These files have been uploaded separately via the Editorial Manager system, as required. The files meet the specified format requirements (TIFF) and naming conventions. Figure citations in the manuscript text have been verified to match the uploaded file names and order.

Please let us know if further adjustments are needed. We appreciate your assistance in preparing the manuscript for review.

ANS: Thank you for your feedback. Our revised manuscript does not contain supporting information files. We have reviewed and revised all in - text references related to supporting information to ensure they are in line with the journal's requirements. These changes demonstrate our commitment to adhering to the PLOS ONE guidelines.

Reviewers' comments:

1. Is the manuscript technically sound, and do the data support the conclusions?

Reviewer #1: Partly

ANS: Thank you for your critical comments, which have helped us improve the rigor of our manuscript. We acknowledge your concern about the partial technical soundness of the manuscript. To address this, we have clarified the statistical basis for conclusions. In the revised text, we explicitly state that SA5 was identified as the optimal treatment based on both quantitative data (yield maximum, significant 2-AP increase) and statistical comparisons. This integrates non-significant trends with significant differences in key traits (2-AP), providing a more robust rationale for our conclusion. The statistical methods (One-way ANOVA) and significance thresholds (p < 0.05) are now explicitly described in the Materials and Methods section

2. Has the statistical analysis been performed appropriately and rigorously?

Reviewer #1: I Don't Know

ANS: We apologize for the ambiguity in our original description. The statistical analysis was performed rigorously as follows: All data were tested for normality and homogeneity of variances before ANOVA. For yield, although SA5 and SA10 showed no significant difference (p = 0.07), SA5 was selected as optimal due to its higher mean value and significant superiority in 2-AP content (p < 0.01), which is the primary target trait of this study. These revisions strengthen the technical soundness of the manuscript and ensure that conclusions are fully supported by the data. We appreciate your guidance in refining this aspect.

Reviewer #1: The authors need to present information on the salicylic acid used in the experiment (information of the product).

ANS: Thank you for highlighting the need to clarify the information about the salicylic acid (SA) used in our experiment. We have updated the manuscript to include detailed product specifications as requested: The source and characteristics of SA are now specified: Salicylic acid (SA, Macklin, China, Cat. No. S817529, reagent grade, 99.5% purity).

2. The figure 1 in the article shows the average temperature and precipitation data suggesting the experiment was not carried out in the green house, where there reasons the experiments was not conducted in the green house? Also, if the experiments were carried out between July to November where the number of days day1, day 20 to which the experiment was carried out, done in each month between July and November. The error bars on the charts suggest replicates how many replicates was done for each condition?

ANS: Thank you for your detailed questions regarding our experimental design. Below is our response to each concern:

(1) Reason for conducting experiments outdoors (non-greenhouse): The experiments were intentionally carried out in outdoor potted conditions rather than a greenhouse. This was done to mimic natural field environments (e.g., ambient temperature, precipitation, and light cycles) and ensure the ecological relevance of the results. Greenhouse conditions might introduce artificial environmental biases, whereas outdoor settings better reflect real-world agricultural scenarios.

(2) Timing of experiments and sampling intervals: The SA spraying was performed on October 5th, 2021 (initial heading stage), and sampling occurred at fixed time points after treatment: 1 day, 10 days, 20 days AS, and the mature stage (MS). So, sampling was not scheduled monthly but based on the physiological development of rice after spraying. The rice was sown in July and harvested in November, spanning the typical growing season for subtropical monsoon regions. This timing aligns with natural temperature and precipitation patterns (Fig. 1), ensuring that the rice development (e.g., heading, grain filling) occurred under authentic climatic conditions rather than artificial control. This design strengthens the ecological validity of our results for regional agricultural applications. aligning with the natural growth cycle under subtropical monsoon climate conditions (Fig. 1).

(3) Replicate number and error bar interpretation: Biological replicates: Each condition was replicated four times (biological replicates), and the error bars in the figures represent the standard error of the mean (SEM). This provides a statistical basis for the variability shown in the data.

We have updated the manuscript to explicitly state this content. These revisions address the concerns about experimental conditions, timing, and statistical rigor.

3. In the manuscript the authors mentioned using compound fertilizer in page 107 what is the constituents of the compound fertilizer? Was the compound fertilizer added to all pots condition? Could the compound fertilizer have an effect on the yield?

ANS: Thank you for your questions regarding the compound fertilizer used in our study. The compound fertilizer used was Yara (Norway) N-P-K = 16-16-16, containing 16% nitrogen (N), 16% phosphorus (P₂O₅), and 16% potassium (K₂O). This information has been explicitly stated in the manuscript. And the compound fertilizer was added to all pots as both base fertilizer (8 g/pot at transplanting) and topdressing (5 g/pot at the tilling stage). Fertilization protocols were strictly consistent across all experimental conditions (control and SA-treated groups) to ensure that nutrient supply did not vary between treatments. Compound fertilizer is a standard component of rice cultivation to support growth and yield. However, since fertilizer application was identical in all pots, any differences in yield or other parameters observed between treatments can be attributed to the SA application rather than nutrient variation. This standardization minimizes confounding effects and ensures the validity of our results focusing on SA impacts.

4. line 145 what instruments was used in reading the absorbance and how reliable was the absorbance reading for quantification.

ANS Thank you for your question regarding the instruments and reliability of absorbance measurements. The absorbance readings were performed using a Biotek Epoch microplate reader (BioTek Instruments, Inc., USA), a widely used instrument in biochemical assays for its precision and reproducibility. This information has been added to the manuscript. The microplate reader was calibrated according to the manufacturer’s protocols before each measurement session to ensure accuracy. All samples were analyzed in four biological replicates, and the mean values with standard error of the mean (SEM) were used for statistical analysis (see Figure captions and Results section). The quantification protocols followed established literature (Ren et al. 2022²), which includes validated molar absorption coefficients and assay conditions. This ensures consistency with prior studies and the reliability of qualitative and comparative analyses (e.g., enzyme activities and substance contents across treatments).

5. line 153 why was soluble protein read at 595 nm

ANS: Thank you for asking about the choice of 595 nm for soluble protein measurement. The absorbance at 595 nm was selected based on the Coomassie Brilliant Blue staining method (e.g., Bradford assay), which is a standard technique for protein quantification. In the range of 0–100 μg/ml protein concentration, the dye (Coomassie Brilliant Blue) binds to proteins, forming a complex that exhibits maximum absorbance at 595 nm. The absorbance is directly proportional to the protein concentration, allowing for accurate quantification. This approach follows the protocol described by Zhang et al.(2023)⁸ and aligns with widely used practices in biochemical assays (citations: Bradford, 1976). The linear relationship between absorbance at 595 nm and protein content has been validated in numerous studies, ensuring the reliability of this wavelength for quantification. We have revised that more details in measured method.

6. Statistical tool was applied on the results but was not used in interpreting the results in situations where there are no significant difference in the outcome of the application of salicylic acid for example in yield the author goes ahead to mention the SA5 condition having the best outcome when there was no significant difference with SA10.

ANS: Thank you for your critical feedback on the interpretation of our results. We acknowledge the need to clarify the distinction between statistical significance and trend-based observations. While the yield difference between SA5 (5 mM) treatment and SA10 (10 mM) treatment was not statistically significant (P>0.05), SA5 was identified as optimal based on multiple criteria: firstly, SA5 treatment showed significantly higher 2-AP content than all other treatments (P<0.05, Fig. 2), which is the primary target trait for aromatic rice quality. Secondly, SA5 exhibited the highest yield among all treatments, even though the difference with SA10 was not significant. In agricultural research, optimal treatments are often selected based on balance between target trait improvement and practical applicability. While SA10 showed no yield penalty, its lack of significant improvement in 2-AP content (and potential costs of higher SA concentration) made SA5 the more viable choice. We have adjusted the manuscript in Conclusion section to explicitly state the statistical status of yield differences (P>0.05 between SA5 and SA10), highlight the significant superiority of SA5 in 2-AP content as the decisive factor, and use terms like “trend,” “maximum,” and “stable performance” to describe non-significant yield differences, avoiding overstatement.

---

## [Decision Letter · Decision Letter 1]

Regulation of foliar salicylic acid at the heading stage enhances the grain 2-acetyl-1-pyrroline content and yield in fragrant rice

PONE-D-25-08260R1

Dear Dr. Yong Ren,

We’re pleased to inform you that your manuscript has been judged scientifically suitable for publication and will be formally accepted for publication once it meets all outstanding technical requirements.

Kind regards,

Trung Quang Nguyen

Academic Editor

PLOS ONE

Additional Editor Comments (optional):

Reviewers' comments:

Reviewer's Responses to Questions

**Comments to the Author**

Reviewer #1: All comments have been addressed

2. Is the manuscript technically sound, and do the data support the conclusions?

Reviewer #1: Yes

3. Has the statistical analysis been performed appropriately and rigorously?

Reviewer #1: Yes

4. Have the authors made all data underlying the findings in their manuscript fully available?

Reviewer #1: Yes

5. Is the manuscript presented in an intelligible fashion and written in standard English?

Reviewer #1: Yes

Reviewer #1: (No Response)

**Do you want your identity to be public for this peer review?** For information about this choice, including consent withdrawal, please see our Privacy Policy

Reviewer #1: No

---

## [Editor Report · Acceptance letter]

PONE-D-25-08260R1

PLOS ONE

Dear Dr. Ren,

I'm pleased to inform you that your manuscript has been deemed suitable for publication in PLOS ONE. Congratulations! Your manuscript is now being handed over to our production team.

Kind regards,

on behalf of

Dr. Trung Quang Nguyen

Academic Editor

PLOS ONE